# The health impacts of a 4-month long community-wide COVID-19 lockdown: Findings from a prospective longitudinal study in the state of Victoria, Australia

Daniel Griffiths[1]*, Luke Sheehan[1], Dennis Petrie[2], Caryn van Vreden[1], Peter Whiteford[3], Alex Collie[1]

1 School of Public Health and Preventive Medicine, Monash University, Melbourne, Australia, 2 Centre for Health Economics, Monash University, Melbourne, Australia, 3 Crawford School of Public Policy, Australian National University, Canberra, Australia

* daniel.griffiths@monash.edu

**Data Availability Statement:** The data are held at Monash University by the Healthy Working Lives Group, School of Public Health and Preventive

## Abstract

### Objectives

To determine health impacts during, and following, an extended community lockdown and COVID-19 outbreak in the Australian state of Victoria, compared with the rest of Australia.

### Methods

A national cohort of 898 working-age Australians enrolled in a longitudinal cohort study, completing surveys before, during, and after a 112-day community lockdown in Victoria (8 July– 27 October 2020). Outcomes included psychological distress, mental and physical health, work, social interactions and finances. Regression models examined health changes during and following lockdown.

### Results

The Victorian lockdown led to increased psychological distress. Health impacts coincided with greater social isolation and work loss. Following the extended lockdown, mental health, work and social interactions recovered to an extent whereby no significant long-lasting effects were identified in Victoria compared to the rest of Australia.

### Conclusion

The Victorian community lockdown had adverse health consequences, which reversed upon release from lockdown. Governments should weigh all potential health impacts of lockdown. Services and programs to reduce the negative impacts of lockdown may include increases in mental health care, encouraging safe social interactions and supports to maintain employment relationships.

Medicine. Data access is restricted by the Monash University Human Research Ethics Committee due to ethical considerations as datasets contain potentially identifying and sensitive information. Procedures to request access to data from this study are available through contacting the Monash University Human Research Ethics Committee at muhrec@monash.edu.

**Funding:** Funding was provided by Monash University and the icare Foundation. The views expressed are those of the authors and may not reflect the views of study funders. Professor Alex Collie is supported by an ARC Future Fellowship (FT190100218). The funders had no role in study design, data collection and analysis, decision to publish, or preparation of the manuscript.

**Competing interests:** The authors have declared that no competing interests exist.

# Introduction

Globally, many public health measures have been employed by governments to reduce viral transmission during the coronavirus pandemic. These include restrictions on gatherings, quarantines, movement and travel restrictions, curfews, business closures, the mandated use of personal protective equipment, mass community infection testing and contact tracing, and more recently vaccination efforts. Responses have varied globally in terms of the stringency of measures imposed, and they have also changed over time [1]. Some of the most stringent restrictions relate to mass quarantines or stay-at-home orders, herein described as *lockdowns*, which have often been coupled with the shutdown of parts of the economy. Lockdowns have been demonstratively effective at reducing the spread of coronavirus [2], although their use has been controversial due to the disruptions they cause to everyday life and adverse health consequences [3]. The World Health Organisation advises the use of lockdowns as short-term measures to regroup, rebalance resources, and protect health workers who are exhausted [4]. Despite this, the use of lockdowns has been widespread globally ranging from 3-day 'snap' or 'circuit-breaker' lockdowns (e.g. in Australia and New Zealand) to extended periods of more than 100 days (e.g. in Argentina, Azerbaijan, Bolivia, Nepal, UK, Peru, Saudi Arabia, Czech Republic, Greece, Germany, Ireland, and Australia). Approximately half of the world's population (3.9 billion people) was under some form of lockdown at one stage in 2020 [5].

There are many known harmful side effects of lockdowns [6]. Stay-at-home orders can bring everyday activities to a standstill, leading to loss of work across non-essential industries, and can result in increased loneliness in the community, or social isolation [7]. Whilst some industries are able to adapt to an increased at-home workforce, many businesses have ceased operations during periods of lockdown. In some countries, financial supports such as wage subsidies were available for some eligible businesses or employees to ensure the maintenance of employment relationship during lockdown periods [8]. Personal finances and social interactions are important determinants of mental health, including during pandemic-induced periods of work cessation [9], and both are likely to be impacted during extended periods of lockdown. In summary, existing evidence suggests that lockdowns will have negative impacts on health, particularly mental health, and on several determinants of health including social interactions, engagement in work and financial resources. To date, very little is known about the persistence of these negative health consequences following the cessation of lockdown. Do communities and individuals recover rapidly as communities re-open, or do the adverse health impacts persist?

Our exploratory analyses focus on a group of working-aged individuals enrolled in a prospective longitudinal study of the health impacts of work loss during the COVID-19 pandemic. Participants completed surveys before, during, and following an extended community lockdown in the Australian state of Victoria [10]. Outcomes in this group were compared to those in a comparison group residing outside of the lockdown area over the same period. By utilising the longitudinal nature of our national dataset, and taking advantage of this unplanned natural experiment, we sought to answer the following research questions:

1. What were the impacts of an extended lockdown related to a COVID-19 outbreak on:

   a. Mental and physical health?

   b. Several determinants of health such as employment, being out of work, social interactions and finances?

2. Are any effects persistent following the conclusion of lockdown?

## Methods

### Setting

Australia experienced two waves of coronavirus during the year 2020. The first smaller wave resulted in a similar set of restrictions across states and territories. The second larger wave was mostly localised to the state of Victoria, and more specifically the metropolitan Melbourne area. The state of Victoria is the second most populous Australian state with 6.68 million people (around 26% of Australia's population), and correspondingly accounts for around a quarter (26.6% during December 2019) of the Australian labour force consisting of 2.90 million Victorian workers [11].

The Victorian state Government response to the second COVID-19 wave included a 112-day extended community lockdown [12]. The nature of the restrictions during the lockdown period changed over time (S1 Table), and between regional and metropolitan areas. During the most stringent period (96.2 / 100 on the government response index [1]), measures included a curfew between 8pm and 5am, a 5-kilometre (i.e. 3 mile) distance limit from home, maximum gathering limits of two people outside the home and no visitors to a home, and mandatory face coverings outdoors and indoors when leaving home [13]. Additionally, there were only four permitted reasons to leave home: shopping for essential items (limited to one person per household), caregiving, or work (where employers were required to support working from home if individuals could work from home), or a maximum of one hour per day of outdoor exercise. A staged easing of restrictions followed the lockdown period (S1 Table). In comparison, for people in the rest of Australia, there were significantly fewer restrictions in place throughout this period, with some differences between state and territory jurisdictions.

### Participants, data and context

We report the findings from a prospective longitudinal Australia-wide cohort study on work and health [9] which was initiated soon after the first wave and, by serendipity, coincided with an extended community lockdown in the Australian state of Victoria. Participants were aged 18 year or older, living in Australia, and were employed in paid work prior to the COVID-19 pandemic. An online Qualtrics survey targeted people that had lost work after the first wave and was promoted via social and general media, and newsletters distributed by community sector and industry groups. The cohort also consisted of participants that completed the survey via a telephone interview conducted by a third-party market research company, and included participants that had lost work, and a group whose working hours had not reduced. Baseline findings from this cohort have been published previously including detail on the recruitment methods and measures [9].

Participants were enrolled in the study and completed a baseline assessment via an online or telephone survey between 27 March to 12 June 2020 [9]. Participants then also completed up to a further three surveys by December 2020, at intervals of one, three and six months following the baseline survey. These follow-up surveys included repeated measures of primary and secondary study outcomes. From these four surveys, responses were transformed into three periods based on the timing of the lockdown in Victoria: (1) *Pre-lockdown*: responses collected prior to 8 July 2020, (2) *Lockdown*: responses collected between 8 July and 27 October 2020 (inclusive), and (3) *Post-lockdown*: responses collected after 27 October 2020. The exposure group comprised people residing in the state Victoria during the lockdown period (Table 1). The comparison group comprised people residing elsewhere in Australia during the lockdown period. Note that the rest of Australia did not experience a lockdown during this period of time. Participants were included in the analysis if they completed at least one survey

**Table 1. Study design demonstrating survey timing and residential location.**

|  | Pre-lockdown | Lockdown (in Victoria) | Post-lockdown |
|---|---|---|---|
| **Exposed Group** (State of Victoria, N = 305) | Data collected 27 March to 7 July 2020 | Data collected 8 July to 27 October 2020 | Data collected 28 October to 31 December 2020 |
| **Comparison Group** (Rest of Australia, N = 593) |  |  |  |

in each of the three periods. For participants completing multiple surveys during the lockdown period, the survey with the most recent data was included to reflect responses that maximised the duration of exposure to the lockdown, and correspondingly surveys completed earlier in the lockdown by the same participant were excluded from analysis. This resulted in three surveys at three-month intervals for each participant. It is important to note that the baseline (pre-lockdown) period coincided with the early stages of the COVID-19 pandemic in Australia, during which significant shifts in health were observed [9].

## Health outcomes

Three health outcomes were assessed. Firstly, the 6-item Kessler Psychological Distress scale was used to evaluate scores of psychological distress ranging from 6 to 30, with moderate to high distress levels ranging from 11 to 30 [14]. The two remaining scores describe mental and physical health and were derived from the respective summary component scores from the 12-item Short-Form health survey (SF-12), where scores of 50 represent pre-pandemic population average, with a standard deviation of 10 [15].

## Determinants of health

Data on several determinants of health were also captured as secondary outcomes. Engagement in work was defined as a binary variable coded from responses describing whether individuals had worked any hours during the prior week, or not. Similarly, a separate binary variable was used to describe whether individuals were either employed or unemployed during each time-point. Two binary measures of social interaction were derived from components of the Social Interaction sub-scale of the Duke Social Support Index describing interactions in the prior week [16]. These included whether people had spent any time with anyone (excluding household members) or not, and whether people engaged in calls (online or telephone) to fewer than seven people during the past week or spoken with seven or more people. The social interaction sub-scale itself was not derived following participant feedback on the question regarding taking part in group meetings (such as sports or other clubs). Many of these activities were not legally permitted under lockdown restrictions and therefore if included the overall interaction sub-scale would instead partially reflect compliance with lockdown restrictions. Financial resources were evaluated with the question *'If all of a sudden you had to get $2000 for something important, could the money be obtained within a week?'* [17]. Responses of 'yes' were categorised as having more financial resources and responses of 'no' and 'don't know' as having less financial resource. Financial stress was measures on a 10-point scale from 1 (not at all stressed) to 10 (as stressed as can be). Financial stress was categorised as responses greater than 5.

## Analytical approach

Descriptive statistics for exposure and comparison groups were calculated. Mean average health outcomes were calculated and graphically represented for the two groups across three

time-points: before, during, and after lockdown. Similarly, group percentages were calculated and graphically represented by location and time-point for the determinants of health.

Mixed linear regression models were used to account for repeated measures of continuous health outcomes with fixed demographic effects. A variable was included describing the timing of lockdown as a random effect using a first-order autoregressive structure with homogeneous variances. The exposure for linear regression models was an interaction term between residential state describing whether individuals resided in Victoria or in the Rest of Australia with a variable describing whether responses were recorded before, during or after lockdown. Fixed effects were included for gender, age group, survey mode, residential location and survey time-point. For models examining the mental health and psychological distress outcomes pre-existing (prior to the first survey) anxiety and depression were also included as fixed effects, and for the physical health outcome model the total number of pre-existing health conditions were included with three categories describing no conditions, one condition and two or more conditions. The reference group was defined as the outcomes after the Victorian lockdown period for individuals living in Australian states and territories other than Victoria.

Generalised estimating equations were used with binary logistic models for binary outcomes describing work status, social isolation, and financial resources, which captured the correlated outcomes within each individual. Each participant was included as a subject variable, and the ternary lockdown context variable was set as a within-subject variable. Models included an interaction term between time-point and location. Main effects were included for gender, age group, survey mode, residential location and survey timepoint.

Statistical tests described as the '*The Lockdown Impact*' were performed for each outcome to investigate pairwise differences for changes in people living in Victoria and people living in the rest of Australia. This analysis aims to describe health impacts of the lockdown whilst accounting for overall changes in health measures over time nationally. More specifically, changes in outcomes were compared between exposed and comparison groups upon two temporal transition categories: (1) *During Lockdown*: differences during the lockdown period controlling for differences pre-lockdown, and (2) *After Lockdown*: post-lockdown differences controlling for pre-lockdown differences.

## Results

A total of 898 participants completed surveys, prior to, during and following the Victorian lockdown period and were included in analyses. There were 161 (18.0%) participants completing two surveys during the lockdown period, and in these cases only the latter survey was included. Cohort demographics and attrition is described and characterised (S2 Table), and all regression models adjust for listed differences between the groups of included or excluded participants. Within the national cohort, 305 (34.0%) participants resided in the state of Victoria with the remaining 593 (66.0%) of participants residing in the rest of Australia and they had similar baseline characteristics (S3 Table).

The Victorian lockdown resulted in significant changes in health (Figs 1 and 2, Table 2), work and determinants of health over time (S1 Fig, S4 Table), and also between individuals in Victoria and the rest of Australia. Amidst relatively high levels of psychological distress in general during the pandemic, distress levels remained elevated for the exposed group during the lockdown period whilst distress levels were lower in the comparison (non-exposed) group. Following the lockdown period, there were no significant differences in psychological distress or mental health of participants in the exposed and comparison groups. During the lockdown period, the mental health of the comparison group significantly improved from pre-lockdown, whereas the no significant differences were observed in the exposure group between pre-

# Psychological distress

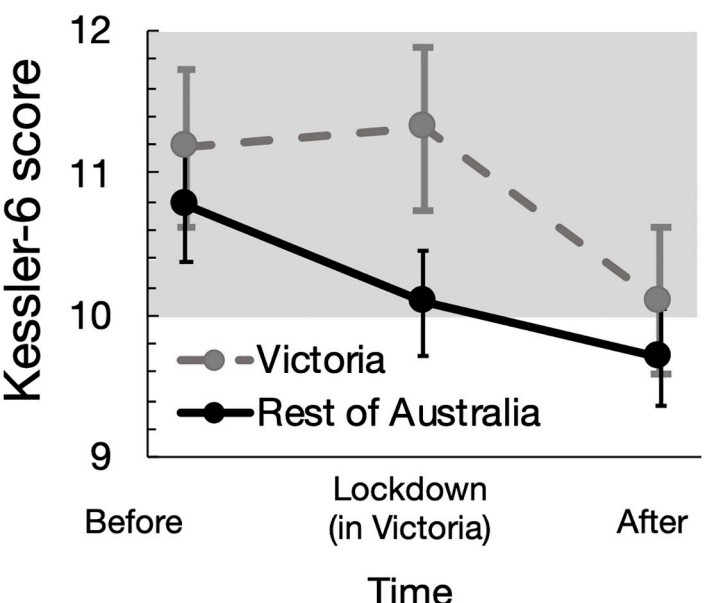

**Fig 1. Changes in psychological distress before, during, and after the community lockdown in Victoria (during 8 July–27 October 2020) compared to the Rest of Australia.** Data describe group mean values and 95% confidence intervals. The shaded region indicates moderate-high distress.

# Mental health

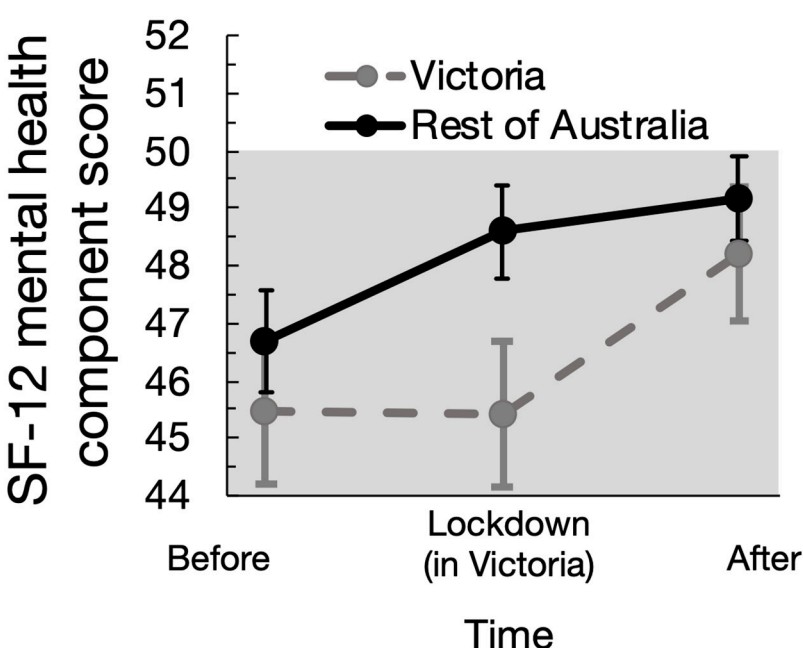

**Fig 2. Changes in mental health before, during, and after the community lockdown in Victoria (during 8 July–27 October 2020) compared to the Rest of Australia.** Data describe group mean values and 95% confidence intervals. The shaded regions indicate below (pre-pandemic) average mental health.

**Table 2. Health impacts of the lockdown in Victoria, Australia.**

| | Adjusted estimates for differences in health score [95% Confidence Interval] | | |
| --- | --- | --- | --- |
| | Psychological distress ($b^* < 0$: worse distress) | Mental health ($b < 0$: poorer health) | Physical health ($b < 0$: poorer health) |
| **The Lockdown Impact** | | | |
| During lockdown | **-0.77** [-1.25–0.29] | **-1.90** [-3.06, -0.74] | 0.04 [-0.82, 0.90] |
| After lockdown | 0.03 [-0.50, 0.56] | 0.27 [-1.03, 1.58] | -0.32 [-1.28, 0.64] |
| **Changes in health over location and time** | | | |
| VIC * pre-lockdown | **-1.30** [-1.85, -0.74] | **-3.23** [-4.48, -1.98] | **1.48** [0.49, 2.47] |
| VIC * lockdown | **-1.36** [-1.92, -0.80] | **-3.23** [-4.48, -1.98] | 0.99 [0.00, 1.98] |
| VIC * post-lockdown | -0.22 [-0.77, 0.34] | -0.48 [-1.73, 0.77] | 0.51 [-0.48, 1.50] |
| RoA * (pre-lockdown) | **-1.05** [-1.36, -0.74] | **-2.48** [-3.24, -1.71] | **0.65** [0.09, 1.21] |
| RoA * (lockdown) | **-0.35** [-0.63, -0.06] | -0.58 [-1.26, 0.10] | 0.12 [-0.38, 0.63] |
| RoA* (post-lockdown) | 0.00 (ref.) | 0.00 (ref.) | 0.00 (ref.) |

Estimates with p < .05 shown in bold.

*Coefficients and confidence intervals listed for psychological distress within the table have been negated (i.e. -b) to ease comparisons with corresponding coefficients for mental health. VIC–the Australian state of Victoria (i.e. lockdown location during 8 July– 27 October 2020). RoA–Rest of Australia (i.e. not Victoria). Models were adjusted for gender, age group, survey mode and by pre-existing medical conditions. *The Lockdown Impact* describes health differences of working-age Victorians to the Rest of Australia, controlling for health differences pre-lockdown.

lockdown and lockdown period. Physical health deteriorated across each time interval nationally, and there were no significant differences identified arising from the lockdown (Table 2, S2 Fig).

Engagement in work (i.e. having worked in the prior week) declined in the exposed group during the lockdown and recovered following the lockdown period. Whilst unemployment levels were significantly higher prior to and during the lockdown nationally compared to after the Victorian lockdown, we did not observe any significant impacts resulting from the lockdown.

During lockdown there was an increase in social isolation, which decreased after the lockdown resulting in no significant post-lockdown effects. Virtual interactions were more common pre-lockdown, and there were no significant changes resulting from the impact of lockdown. Financial stress and people with fewer financial resources decreased over time in the cohort nationally, and there were no significant differences resulting from Victoria's lockdown.

## Discussion

Our findings demonstrate that the extended community lockdown in the state of Victoria during the southern hemisphere winter and spring of 2020 had negative mental health consequences for people of working age including increases in psychological distress as assessed using the Kessler-6 scale, and decreases in mental health assessed using the SF-12 mental component score. However, these effects were short-lived. Following an abrupt national decline in mental health observed during the early stages of the pandemic, people exposed to an extended lockdown experienced a delay in the recovery of mental health observed in the rest of Australia. Following the easing of lockdown measures, we then observe a resolution of the negative mental health impacts of lockdown within a two-month period, bringing the mental health of Victorian residents back in line with that of the rest of Australia. We also observed a pattern of

deterioration during lockdown, followed by recovery post lockdown, in two determinants of health: social interactions and engagement in paid work. Whilst nationally we observed significant reductions in financial stress, physical health and unemployment throughout the pandemic period, along with increases in virtual interactions with others, significant impacts of the extended Victorian lockdown were not elucidated.

The negative psychological consequences of enforced quarantine are well described [18]. A recent meta-analysis of psychological health in COVID-19 induced lockdowns reported an overall small effect on mental health symptoms among those exposed to lockdown conditions, with substantial heterogeneity among studies [19]. The studies in this review reported lockdown periods ranging from 1 to 60 days, and included a range of study designs including comparisons between groups exposed and not exposed to lockdowns, or within person designs comparing mental health before and after exposure to lockdown orders. Our study utilises both a within subject (pre, during and post-lockdown) measurement and incorporates a contemporaneous comparison group also assessed longitudinally. The duration of exposure to lockdown conditions in our study is also much longer than those reported in this meta-analysis, at 112 days. We also identified changes in several determinants of health occurring during the lockdown period, including a reduction in social interactions and engagement in work, both of which may have contributed to the increased psychological distress and poorer mental health observed.

A number of strategies for mitigating the negative psychological consequences of lockdown have been proposed [18]. These include limiting the duration of quarantine restrictions, effective public health messaging, ensuring ongoing access to basic supplies, and actions that enable social interaction [18]. In the state of Victoria during the 2020 lockdown studied in this paper, the official public health messaging occurred on an ongoing basis via daily press conferences with the Premier and senior government health officials, and release of information and statistics via social media and health department websites. One feature of the Victorian lockdown was regular changes in the restrictions being enforced. For example, initially the lockdown rules allowed exemptions to movement restrictions for visiting intimate partners [20], and this was later extended to include other forms of social bubbles for people living alone [21]. These regular changes may have contributed to confusion among those required to enforce compulsory restrictions [22], and among members of the public.

The burden of mental ill health was apparent from the early stages of the COVID-19 pandemic, and prior to the Victorian lockdown period, demonstrating the need for additional mental health supports and services during the pandemic [9,22,23]. Our findings suggest that during periods of extended lockdown this need becomes more acute, coinciding with increased loss of work and social isolation. As a result, dedicated mental health supports and services are encouraged during periods of lockdown, in addition to actions that are anticipated to ameliorate some of the negative impacts of lockdowns on determinants of health such as minimising loss of work and reducing loneliness.

Financial loss has been highlighted as a potential post-lockdown stressor [18]. Our findings demonstrated that financial stress was highest during the early stages of the pandemic, and decreased over time nationally. Furthermore, we did not identify any significant differences in financial stress due to the lockdown within our sample. This may be due, in part, to several financial supports for businesses suffering significant losses or closures during the Victorian lockdown such as dedicated lump sum payments [24], and extended eligibility for a national wage subsidy program due to the lockdown measures [25]. Nationally, economic support for households and businesses was estimated to amount to around 15% of GDP [26], of which wage subsidies and temporary doubling of payments to income support recipients (e.g. unemployment allowance) constituted 6% of GDP [27]. These supports are anticipated to have

contributed to reduced financial stress for people experiencing loss of work due to the lockdown, and consequently ameliorated further deteriorations in mental health.

The Victorian lockdown shaped the nature of work over this period. Many businesses had to temporarily pause operations, whilst others adapted to working from home arrangements in line with Government directives [12]. Our findings suggest that engagement in work increased nationally from the early stages of the pandemic, and that this increase was hampered during the lockdown period, yet recovered thereafter. Given the decrease in work over the lockdown period, encouraging businesses to develop pandemic response plans [28] or operating models that incorporate adaptive work options for employees during periods of lockdown are likely to support the mental health of employees. Some changes to work, such as increased working from home arrangements for people in Victoria, are expected to have long-lasting effects over the coming years. [29] The health impacts of increased working from home arrangements in the workforce resulting from lockdowns, and the pandemic more generally, are yet to be fully uncovered, and warrant further longitudinal research.

Due to our focus on individuals experiencing loss of work during the pandemic, several demographic groups are either underrepresented or absent in our sample, including older adults, children and people that were not engaged in paid work prior to the pandemic. It has been recognised that the experiences of lockdown by these groups will differ due to impacts of school closures resulting in debilitating effects on children's development, education and behavioural problems [30], and disproportionate consequences for older adults whose primary social contact is outside the home [31]. Whilst our sample of individuals is not nationally representative, we accounted for age-related and gender-specific differences within the working-age cohort. Additionally, the use of a comparison group assisted to account for other forms of bias, such as the differences in cohort retention based on survey mode. However, this comparative approach has limitations for forms of attrition bias resulting changes in health due to the experience of lockdown, such as increased attrition for individuals in distress. The risk of the Victorian COVID-19 outbreak spreading elsewhere nationally may have contributed to negative shifts mental health for people in the rest of Australia. In such circumstances, our findings may be underreporting the magnitude of mental health consequences, and of determinants of health, during lockdowns.

While our findings describe group-level changes in day-to-day psychological distress and several determinants of health during and after lockdown, the impacts of lockdown are unlikely to be shared equally. We have previously reported on the negative mental health impacts of pandemic-related loss of work, and how individuals with pre-existing mental health conditions, or those with fewer social or financial supports, are most at-risk of poor mental health [9]. Additional supports are warranted for these groups upon exposure to lockdown measures. We were limited in our ability to investigate the health consequences of lockdowns in other potentially vulnerable groups such as disadvantaged communities, Aboriginal and Torres Strait Islanders or culturally and linguistically diverse communities due to their small sample size. It is noted that people who experienced elevated levels of anxiety either before or after lockdown report symptoms of poor physical health [32], although this was not explored within the context of our analysis. Whilst our findings describe a recovery in aspects of mental health such as symptoms of anxiety and psychological distress, mental disorders such as post-traumatic stress disorder may re-emerge over longer periods of time [33], or upon repeated exposure to lockdowns or crises.

The nature of lockdowns has substantially differed globally, in terms of the stringency of restrictions, durations, the repeated nature of lockdowns and government responses to them [1]. Our findings describe the health impacts during and after a stringent 112-day community lockdown in Victoria, Australia. However, the experiences we have observed may vary in

other contexts. Thus, a need for similar studies across a range of scenarios and contexts is required to truly understand the impact of lockdowns on health and its determinants.

## Conclusion

The extended, 112-day community lockdown in the state of Victoria during the winter and early spring of 2020 had a negative impact on mental health, engagement in work and social interactions. After the removal of lockdown restrictions, the mental health of those exposed to the extended lockdown returned to be equivalent to those in the non-exposed comparison group. Restrictions that promote safe forms of social interactions and engagements in work may mitigate some negative psychological consequences during periods of lockdown. Governments and employers can help to minimise the negative mental health effects of lockdowns by taking actions and implementing policy that encourages maintenance of employment relationships, supports engagement in work and promotes social interaction and addresses the mental health needs of the community. Longer-term follow-up is required to identify any longer-term persistent health effects of community lockdowns.

## Supporting information

**S1 Fig. Changes in determinants of health prior to, during, and after the community lockdown in Victoria, compared to the Rest of Australia.**
(DOCX)

**S2 Fig. Changes in physical health prior to, during, and after the community lockdown in Victoria compared to the Rest of Australia.**
(DOCX)

**S1 Table. Summary of restrictions leading to, during, and following the 2020 extended community lockdown in Victoria, Australia.**
(DOCX)

**S2 Table. Comparisons of participants included in the analyses and those excluded.**
(DOCX)

**S3 Table. Comparisons of participants residing in Victoria compared to participants in the Rest of Australia.**
(DOCX)

**S4 Table. Impacts of the lockdown in Victoria on determinants of health including work, social interactions, and finance.**
(DOCX)

**S1 Appendix. Media statements and Victorian Government publications on restrictions.**
(DOCX)

## Acknowledgments

We acknowledge the Social Research Centre for undertaking telephone interviews. We thank the participants for taking the time to complete the surveys.

## Declarations

### Ethics approval and consent to participate

Approval to conduct the study was provided by Monash University Human Research Ethics Committee (#24003). All procedures followed were in accordance with the ethical standards of the responsible committee on human experimentation (institutional and national) and with the Helsinki Declaration of 1975, as revised in 2000 (5). Informed consent was obtained from all patients for being included in the study.

## Author Contributions

**Conceptualization:** Alex Collie.

**Data curation:** Daniel Griffiths, Luke Sheehan, Caryn van Vreden.

**Formal analysis:** Daniel Griffiths.

**Funding acquisition:** Alex Collie.

**Investigation:** Daniel Griffiths, Luke Sheehan, Dennis Petrie, Caryn van Vreden, Peter Whiteford, Alex Collie.

**Methodology:** Daniel Griffiths, Luke Sheehan, Dennis Petrie, Alex Collie.

**Project administration:** Luke Sheehan, Caryn van Vreden, Alex Collie.

**Supervision:** Alex Collie.

**Writing – original draft:** Daniel Griffiths.

**Writing – review & editing:** Daniel Griffiths, Luke Sheehan, Dennis Petrie, Caryn van Vreden, Peter Whiteford, Alex Collie.

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
