## [Decision Letter · Decision Letter 0]

11 Oct 2021

PONE-D-21-24391The health impacts of a 4-month long community-wide COVID-19 lockdown: Findings from a prospective longitudinal study in the state of Victoria, Australia.PLOS ONE

Dear Dr. Griffiths,

Thank you for submitting your manuscript to PLOS ONE. After careful consideration, we feel that it has merit but does not fully meet PLOS ONE’s publication criteria as it currently stands. Therefore, we invite you to submit a revised version of the manuscript that addresses the points raised during the review process.

 I ask that you first - that is, prior to attempting revisions - please review the comment provided by Reviewer 5 regarding a similarly titled paper. As the Reviewer notes, this may well be a preprint or conference paper. Could you please first review this comment and provide in correspondence with me a response to that query. Following resolution of that query, please consider submitting revisions in line with those outlined by reviewers 2, 3 and 4 in whole. Please consider the revisions and comments specified by those reviewers in light of each other - and make a holistic response to them at appropriate moments. 

Reviewer 1 and reviewer 5 provide more substantial critique of the paper and its methods. Please review these comments and respond to each point raised in a neatly set out table (with verbatim extract of the relevant reviewer comment) appended in a separate document (your rebuttal letter) with a brief comment against each query raised with reference to the areas (or main areas) where you have made edits in response. Please also ensure that any edits to the manuscript are provided in mark up for review.

I note that some of the comments made by Reviewer 1 and 5 may well be similar in nature - so please respond in line line of the table to both. Whilst Reviewer 5 has recommended a rejection, given the commentary by other reviewers and my own views, I believe an opportunity to make revisions to the manuscript is warranted.

We look forward to receiving your revised manuscript.

Kind regards,

David James Carter, PhD, LLM (Res), LLB (Hon I), BA

Academic Editor

PLOS ONE

3. Please state whether you validated the questionnaire prior to testing on study participants. Please provide details regarding the validation group within the methods section.

Reviewers' comments:

Reviewer's Responses to Questions

**Comments to the Author**

1. Is the manuscript technically sound, and do the data support the conclusions?

Reviewer #1: Partly

Reviewer #2: Partly

Reviewer #3: Partly

Reviewer #4: Yes

Reviewer #5: No

2. Has the statistical analysis been performed appropriately and rigorously? 

Reviewer #1: No

Reviewer #2: Yes

Reviewer #3: I Don't Know

Reviewer #4: I Don't Know

Reviewer #5: No

3. Have the authors made all data underlying the findings in their manuscript fully available?

Reviewer #1: No

Reviewer #2: Yes

Reviewer #3: Yes

Reviewer #4: Yes

Reviewer #5: No

4. Is the manuscript presented in an intelligible fashion and written in standard English?

Reviewer #1: Yes

Reviewer #2: Yes

Reviewer #3: Yes

Reviewer #4: Yes

Reviewer #5: Yes

5. Review Comments to the Author

Reviewer #1: Summary:

This paper summarized a survey to find the association of lockdown with psychological distress in Victorian community in Australia. N=898 subjects who finished the survey in all time points was included in the study. The paper contributes to understanding people’s psychological distress influenced by the lockdown policy, but recovered after the lockdown. This paper is practical and meaningful.

The topic of this study is interesting. It helped us to understand how the lockdown policy’s influence on people’s mental health. The survey monitored people’s status longitudinally across the time points. This paper was well written and structured logically. It clearly introduce the background and findings in the discussion.

Concerns:

There are quite a few concerns on methods and analysis and unclear points in this paper. There are some major issues:

1. The data collection background was not clearly explained. The survey Reponses were from the working age group, so the population is the entire working-age group in corresponding areas/community, but what is the population of the study subjects? What is the exact definition of working-age group?

From the supplementary table S1, the retention group N=898 contains subjects with age 65 and more. So, more clarification and discussion are necessary.

2. How was the survey sample size decided? There is no power analysis discussion in the paper. What is the power or effect size from the current sample?

3. The data collection process was confusing. How the survey participants were recruited? Randomly selection? Volunteer response? Response rate? More explanation is needed.

4. There is only a rough descriptive table between sample and subjects excluded from the study. But, there is no descriptive tables to compare exposure group (N=305) and Comparison group (N=593) on age, gender, etc. it is concern that if there were unbalance between these two groups.

5. From table S2, the descriptive table between Retention group and attribution group (excluded subjects), it seems the retention group is mainly telephone interview (80%), and attribution group were more taking online survey (60%). I think there is bias. Revision and More discussion is needed

6. In terms of the main finding, for example the figure of mental health, the conference intervals are needed to show the variation of each group across different time points (before, lockdown and After). What is the effect size of this differences claimed in the paper?

7. Also, sensitivity analysis needs to be added to show the robustness of the finding.

Reviewer #2: Thank you for the opportunity to review such a beautifully written paper on a topic of great national importance.

I would recommend acceptance if one issue can be remedied.

The study evaluates mental health at different time points and finds no lasting effects from the lockdown. To reach this conclusion it used the Kessler 6 scale and the mental component of the SF-12.

My concern is that the scales used may not capture the full range of mental health impacts of lockdown. For instance, this range includes disorders that have a complicated aetiology, such as post-traumatic stress disorder (PTSD), where symptoms may emerge over time. In fact, in the literature on disaster response suggests there may be a 'honeymoon phase' characterised by increased community cohesion in the immediate aftermath of a mass trauma event (see e.g. DeWolfe, 2000, p.5). Post-traumatic stress is one of the most likely sequelae of the crisis, yet I can find no research on the validity of the SF-12 or the K-6 for discerning PTSD symptoms in a study population.

I would request the manuscript be amended to account for these limitations. It would be more accurate to say, hewing closer to the wording of the scales themselves, that day-to-day stress and distress returned to pre-lockdown levels, but impacts on the longer-term mental health of participants cannot be assessed using the methodology adopted in the study.

----

DeWolfe 2000 https://www.hsdl.org/?view&did=4017 accessed 13 Sept 2021.

Reviewer #3: Your paper is both timely and adds to our knowledge of the impact of lockdowns on the mental health of the population. I believe your paper could be strengthened with more detail - at the moment, yes it substantiates the fact that people in lockdown in Victoria suffered from worse mental health, but we don't know much about the nuances therein. I note that you have not analysed the experiences of your subjects based on whether they are Aboriginal or Torres Strait Islanders or from CALD backgrounds, or for example where they lived specifically, and even though you note that you looked a gender, you did not discuss any differences in detail.

Reviewer #4: This well-written and timely paper investigates the short- and long-term negative effects of lockdown in the State of Victoria, Australia. The authors found that the Victorian lockdown led to increased psychological distress during the lockdown, but these effects resolved after restrictions are lifted.

The study raises important points regarding the negative effects during and following lockdowns and contributes to existing works on this important topic.

However, the persistence of the negative health effects needed to be evaluated more carefully. Only showing that negative mental health consequences resolve over time is too simplistic. Since the Covid-19 pandemic began, we have learned that lockdown and social isolation measures have different effects for different populations. Some communities are more vulnerable than others. For example, disadvantaged low-income communities and people with mental health and addiction problems are more susceptible to negative impacts from interventions such as lockdowns. It would be useful for the author to provide information about the impacts of these measures on vulnerable individuals during and following lockdowns. If the authors don’t have the data regarding the effects on these populations, they should acknowledge and discuss this limitation.

The same applies to the discussion around the Australian government’s efforts to mitigate the harm caused by public health measures against Covid-19. Discussion is needed on the strategies for mitigating the negative psychological consequences of lockdown specifically for populations more susceptible to public health measures. Can the author also discuss strategies to assist vulnerable communities during lockdowns and relate these to the situation in Victoria?

Last, there would be value in the author adding a paragraph comparing their findings to similar studies conducted in other countries (e.g., the short-/long-term negative effects of lockdowns) and discuss variation in outcomes (if such exists).

Reviewer #5: Comments to Author:

Ms. Ref. No. PONE-D-21-24391

Title: The health impacts of a 4-month long community-wide COVID-19 lockdown: Findings from a prospective longitudinal study in the state of Victoria, Australia.

• Overall objective and the extent to which this was achieved:

This paper attempts to address a very important and, to date, a sparsely researched topic: When lockdown ceases, to what extent do the established negative health effects of lockdown persist and what is the speed of recovery of individuals and communities to these negative effects?

The authors state that they took advantage of a natural experiment where, contemporaneously, there was a comparison group outside the lockdown area in Victoria, Australia.

Their stated aim was to examine (lines 83-87):

1. the impact on mental and physical health of the extended lockdown.

2. the impact of the extended lockdown on established determinants of health: unemployment, social interaction, and finances.

3. Effects persisting after lockdown was lifted.

Of these three aims, the literature is flooded with the impact of lockdowns on health, mental, physical, and psychological. This is also mentioned in their introduction. However, it is the third aspect that is under-researched as again, they explain (lines 70 -73) even before the three objectives are listed.

Main takeaway: Unfortunately, it is in this third objective that the paper falls short. In their conclusion, they mainly talk about the negative impact of the extended lockdown (not new ground) and the mitigating effect of employment and social interactions which also has been shown. They end by saying “longer-term follow-up is required to identify any persistent health effects of community lockdowns.” They actually could have done this had they exploited the 2 months of post-lockdown data. But that was not done.

• Clarification needed: The paper seems remarkably similar to this already published:

Griffiths, D., Sheehan, L., Vreden, C.V., Petrie, D., Sim, M. and Collie, A., 2021. 1360 Health impacts of a 4 month community-wide lockdown: a prospective longitudinal study in Victoria, Australia. International Journal of Epidemiology, 50(Supplement_1), pp.dyab168-240.

Though this appears in Google Scholar, do I assume that this is just the publication of an abstract of a conference presentation? I was unable to get hold of this paper. The authors can clarify.

• The Model:

1. I failed to find out what is the model that they are estimating. Lines 170 to 189 give a general description of what was done. The exact specification of the model/models needs to be spelled out, and the control variables indicated. This is the heart of the paper and a general description does not provide the information needed to assess the underpinnings of the analysis

2. In the estimation, they also need to add a fixed time effect.

• Variables:

3. What were all the variables and how were they defined and measured? For instance, I could find no definition of the two key variables they have used and even shown graphs with them: psychological distress and mental health.

4. Of the variables they have discussed, I have the following comments:

i) Financial stress is important to capture. However, the authors have taken only one out of the 3 indicators that the GSS Summary Results for Australia have considered. The GSS considers three elements:

(https://www.abs.gov.au/statistics/people/people-and-communities/general-social-survey-summary-results-australia/2014)

a. A cash flow problem in the last year in terms of either inability to pay bills on time or having to get help to pay.

b. The ability to raise $2000 within a week for something important.

c. Taking a dissaving action such as “drawing on savings, increasing a credit card balance by $1,000 or more and taking out a personal loan”.

Please see Graph 2 in https://www.abs.gov.au/statistics/people/people-and-communities/general-social-survey-summary-results-australia/2014. It can be seen that of all the 3 stressors, cash flow and dissaving have higher percentages in general than the ability to raise $2000 dollars. In fact, it is the unemployed that have the most problems.

The authors need to justify why the other two were not considered and why focusing on the selected item was appropriate.

ii) I do not understand how, (if the responses to that one question was taken as “yes” or “no” or “don’t know”), financial stress could be measured (for each person) on a 1 to 10 scale. Is this an overall (average) measure for the county? This needs clarification.

5. Social Interaction – two binary measures are taken:

Was any time spent with anyone not a household member and

Whether contact was made with 7 or fewer people via speech, phone or online communication.

The original 35 item Duke Social Support Index was abbreviated to a 23-item index and also to a 11-item index for use on elderly patients who may get exhausted by the use of the longer questionnaire. However, studies have generally used either the 23 item or the 11 item DSSI. Usually these have been for older people. In this paper (not confined only to the elderly), only two items have been picked and the justification of these two items has not been given. In other words, why would we believe in the validity of a 2-item DSSI?

The authors need to justify and discuss this.

6. What is the need for the variable “engagement in work” since “employment” at time of survey is also included? Is there any extra information being captured?

• Other comments regarding results:

Turning to the results with regard to the stated objectives:

7. The negative health findings (they focus on psychological distress and mental health --- again, how were these measured?) during lockdown for both exposed and comparison group is in line with findings elsewhere in the COVID literature and so do not break new ground.

• Unobserved heterogeneities as the two groups do not appear similar

8. During the lockdown period the mental health of the comparison group actually improved significantly from pre-lockdown. There is no further exploration of this counter-intuitive result. However, this pattern did not show up in the exposure group thus leading one to suspect that there are other factors at play or unobserved heterogeneities that have not been captured and which may, thus, invalidate this “natural experiment”.

9. Right now, looking at the mental health and psychological graphs, it can be seen that the two groups did not begin at the same level. Are these differences statistically significant? This is important as otherwise, this further shows that the two groups were quite different to begin with (another reason to suspect other unaccounted for factors at play).

10. What accounts for the different trajectories of psychological distress? Again, the two groups appear different.

• The graphs

11. The graphs are not easy to understand.

i. Since there is no discussion of “health score” in the cases of mental health and psychological distress, one does not know how to interpret the y-axes. The x-axis is not labelled.

ii. How were the health scores obtained?

iii. As I understand, the first dot is for data collected PRIOR to lockdown. The second dot is the data collected DURING lockdown. The third dot is the data collected POST lockdown. It seems a valuable opportunity to explore the POST lockdown period was not taken by averaging the October to December data. This would have shown the speed of recovery and might have added some new information to the literature.

• Precision of estimates and statistical power:

12. Since I am not clear on the models estimated to find the mental and physical effects of a lockdown, I wanted to make sure that if any of the explanatory variables were correlated, that was clearly brought out and the implications mentioned. Employment, social interaction, and financial stress are likely to be highly correlated. Though there are methods to deal with this problem, at least the impact on the precision of the estimates needs to be mentioned.

13. The state of Victoria has 6.68 million people (line 94) accounting for over a quarter of the 6.68 million population of Australia (line 95). I find it worrying that the analysis has only 305 people in Victoria (exposed group) and 593 in the rest of Australia (the comparison group). The sample sizes seem too small for the statistical tests to have much power. At the very least, this should be mentioned.

6. PLOS authors have the option to publish the peer review history of their article (what does this mean?). If published, this will include your full peer review and any attached files.

Reviewer #1: No

Reviewer #2: No

Reviewer #3: No

Reviewer #4: No

Reviewer #5: No

---

## [Author Response · Author response to Decision Letter 0]

14 Dec 2021

The authors have addressed each of the comments from the 5 reviewers in full. This documented in a Table, and also repeated here below.

Responses to reviewer comments

Manuscript ID: PONE-D-21-24391

Title: The health impacts of a 4-month long community-wide COVID-19 lockdown: Findings from a prospective longitudinal study in the state of Victoria, Australia.

We thank the reviewers for their consideration of this paper and comments. Our responses are listed in the table below.

REVIEWER 1:

This paper summarized a survey to find the association of lockdown with psychological distress in Victorian community in Australia. N=898 subjects who finished the survey in all time points was included in the study. The paper contributes to understanding people’s psychological distress influenced by the lockdown policy, but recovered after the lockdown. This paper is practical and meaningful.

The topic of this study is interesting. It helped us to understand how the lockdown policy’s influence on people’s mental health. The survey monitored people’s status longitudinally across the time points. This paper was well written and structured logically. It clearly introduce the background and findings in the discussion.

REVIEWER COMMENT:

Concerns:

There are quite a few concerns on methods and analysis and unclear points in this paper. There are some major issues:

1. The data collection background was not clearly explained. The survey Reponses were from the working age group, so the population is the entire working-age group in corresponding areas/community, but what is the population of the study subjects? What is the exact definition of working-age group?

From the supplementary table S1, the retention group N=898 contains subjects with age 65 and more. So, more clarification and discussion are necessary.

AUTHOR RESPONSE:

Text has been added outlining the inclusion criteria for the study which includes being aged 18 years or more, and were working before the pandemic. The recruitment process is now outlined in more detail.

Paragraph one has been added to the section:

Methods>Participants, data and context 

REVIEWER COMMENT:

2. How was the survey sample size decided? There is no power analysis discussion in the paper. What is the power or effect size from the current sample? 

AUTHOR RESPONSE:

A power analysis was included for primary study outcomes for the longitudinal cohort study. However, this paper describes an unplanned experiment looking at the impacts of a lockdown that took place within the study period, thus we had no opportunity to choose (i.e. increase) the sample size at this stage and a power analysis was not conducted to observe the health impacts during, and after, an unanticipated community lockdown.

The sizes of effects are described in Table 2 in the forms of adjusted estimates on the scales of either differences in scores on the Kessler-6 scale or the SF-12 mental or physical component summary scale. Effect sizes range from medium to low depending on the outcome measure and the confidence intervals demonstrate, given our sample size, the statistical power we have for inference.

REVIEWER COMMENT:

3. The data collection process was confusing. How the survey participants were recruited? Randomly selection? Volunteer response? Response rate? More explanation is needed. 

AUTHOR RESPONSE:

We have added details within this paper explaining the recruitment process. This is also described in reference 9.

Paragraph one has been added to the section:

Methods>Participants, data and context 

REVIEWER COMMENT:

4. There is only a rough descriptive table between sample and subjects excluded from the study. But, there is no descriptive tables to compare exposure group (N=305) and Comparison group (N=593) on age, gender, etc. it is concern that if there were unbalance between these two groups.

AUTHOR RESPONSE:

Supplementary Table S3 has been added comparing descriptive statistics of the exposure group and comparison group. The groups are largely similar and the reported variables in this Table are incorporated in the multivariate regression models to account for any imbalance between groups.

REVIEWER COMMENT:

5. From table S2, the descriptive table between Retention group and attribution group (excluded subjects), it seems the retention group is mainly telephone interview (80%), and attribution group were more taking online survey (60%). I think there is bias. Revision and More discussion is needed. 

AUTHOR RESPONSE:

This comment now refers to Table S3 since the addition of an extra supplementary table.

The comparison throughout this paper is between the exposure group and control group which have very similar distributions for the survey mode. The new Supplementary Table S2 has been added following a reviewer comment to illustrate this. 

Whilst there is a clear difference in attrition based on the survey mode, both groups share the same level of attrition. Furthermore, survey mode is included as a covariate in all models. 

The following sentence has been added to the Discussion:

“the use of a comparison group assisted to account for other forms of bias, such as the differences in cohort retention based on survey mode”

REVIEWER COMMENT:

6. In terms of the main finding, for example the figure of mental health, the conference intervals are needed to show the variation of each group across different time points (before, lockdown and After). What is the effect size of this differences claimed in the paper? 

AUTHOR RESPONSE:

Figures on health outcomes, and those on determinants of health have been adapted to include an estimate of the uncertainty within each group at each time point using 95% confidence intervals.

Effect sizes (and respective 95% CIs) are explicitly stated in Table 1, (and Supplementary Table S4) in the form of changes in scores to either the Kessler-6 scale or the SF-12 component summary score, (and group percentage changes in determinants of health). Note that the uncertainty in the average change in scores (tables) may be smaller than the uncertainty at each point in time (Figures) due to the within individual correlation in scores over time.

REVIEWER COMMENT:

7. Also, sensitivity analysis needs to be added to show the robustness of the finding. 

AUTHOR RESPONSE:

While we do not refer to an explicit ‘sensitivity analysis’, the figures represent the unadjusted estimates and the regression results the adjusted estimated (and both illustrate the same story). In addition, the regression models have been replicated on 9 occasions across 4 themes (health, work, social interactions and finance).

For example, the robustness of the findings regarding aspects of mental health can be observed by similar patterns in mental health SF-12 scores and (inverted) Kessler-6 psychological distress scores. Changes observed in mental health and distress do not extend to health more generally such as physical health (based on the measures used).

Similarly, the regression model for social isolation demonstrates a significant effect of lockdown, and recovery thereafter, whereas the lack of an observed significant effect for virtual interactions acts to demonstrate an element of robustness.

Analogous considerations can be inferred by comparing the outcomes for engagement in work vs. employment, and financial stress vs. financial resources.

Taken together, this approach demonstrates where key differences are observed during lockdown across the aforementioned four themes, and their recovery thereafter. It also demonstrates outcomes where significant effects are not observed.

REVIEWER 2:

Thank you for the opportunity to review such a beautifully written paper on a topic of great national importance.

I would recommend acceptance if one issue can be remedied.

REVIEWER COMMENT:

The study evaluates mental health at different time points and finds no lasting effects from the lockdown. To reach this conclusion it used the Kessler 6 scale and the mental component of the SF-12.

My concern is that the scales used may not capture the full range of mental health impacts of lockdown. For instance, this range includes disorders that have a complicated aetiology, such as post-traumatic stress disorder (PTSD), where symptoms may emerge over time. In fact, in the literature on disaster response suggests there may be a 'honeymoon phase' characterised by increased community cohesion in the immediate aftermath of a mass trauma event (see e.g. DeWolfe, 2000, p.5). Post-traumatic stress is one of the most likely sequelae of the crisis, yet I can find no research on the validity of the SF-12 or the K-6 for discerning PTSD symptoms in a study population.

I would request the manuscript be amended to account for these limitations. It would be more accurate to say, hewing closer to the wording of the scales themselves, that day-to-day stress and distress returned to pre-lockdown levels, but impacts on the longer-term mental health of participants cannot be assessed using the methodology adopted in the study. 

DeWolfe 2000 https://www.hsdl.org/?view&did=4017 accessed 13 Sept 2021.

AUTHOR RESPONSE:

The reviewer raises an important point on capturing the full range of mental health impacts and the scope of the mental health measures used in the study.

A penultimate paragraph has been added to the Discussion section addressing the points raised by the reviewer.

The phases of disaster described by the DeWolfe Training manual is relevant and has been cited.

REVIEWER 3

Your paper is both timely and adds to our knowledge of the impact of lockdowns on the mental health of the population.

REVIEWER COMMENT:

I believe your paper could be strengthened with more detail - at the moment, yes it substantiates the fact that people in lockdown in Victoria suffered from worse mental health, but we don't know much about the nuances therein. I note that you have not analysed the experiences of your subjects based on whether they are Aboriginal or Torres Strait Islanders or from CALD backgrounds, or for example where they lived specifically, and even though you note that you looked a gender, you did not discuss any differences in detail. 

AUTHOR RESPONSE:

While we would have loved to explore further in terms of how the lockdowns impacted different groups in society we were constrained by our sample size to be able to provide robust estimates for these different groups.. 

We now more clearly acknowledge that the nuances of changes in mental health across different groups of people are not described in this paper. 

We reference a paper looking at the risk of poor mental health based on several determinants of health and gender-differences [9].

A penultimate paragraph has been added to the Discussion section addressing the points raised by the reviewer, including our limited sample of Aboriginal and Torres Strat Islanders and CALD communities.

REVIEWER 4:

This well-written and timely paper investigates the short- and long-term negative effects of lockdown in the State of Victoria, Australia. The authors found that the Victorian lockdown led to increased psychological distress during the lockdown, but these effects resolved after restrictions are lifted.

The study raises important points regarding the negative effects during and following lockdowns and contributes to existing works on this important topic.

REVIEWER COMMENT:

However, the persistence of the negative health effects needed to be evaluated more carefully. Only showing that negative mental health consequences resolve over time is too simplistic. Since the Covid-19 pandemic began, we have learned that lockdown and social isolation measures have different effects for different populations. Some communities are more vulnerable than others. For example, disadvantaged low-income communities and people with mental health and addiction problems are more susceptible to negative impacts from interventions such as lockdowns. It would be useful for the author to provide information about the impacts of these measures on vulnerable individuals during and following lockdowns. If the authors don’t have the data regarding the effects on these populations, they should acknowledge and discuss this limitation. 

AUTHOR RESPONSE:

A similar point has also been described by reviewer 3. 

A penultimate paragraph has been added to the Discussion section addressing the points raised by the reviewer. 

We did not collect data on addiction, but reference a paper describing the role of pre-existing mental health conditions on changes in mental health upon loss of work during the pandemic [9]. 

REVIEWER COMMENT:

The same applies to the discussion around the Australian government’s efforts to mitigate the harm caused by public health measures against Covid-19. Discussion is needed on the strategies for mitigating the negative psychological consequences of lockdown specifically for populations more susceptible to public health measures. Can the author also discuss strategies to assist vulnerable communities during lockdowns and relate these to the situation in Victoria? 

AUTHOR RESPONSE: 

These are important points, although we have little information on this resulting from the reported analysis in this paper for the cohort.

A penultimate paragraph has been added to the Discussion section briefly covering the points raised by the reviewer

REVIEWER COMMENT: 

Last, there would be value in the author adding a paragraph comparing their findings to similar studies conducted in other countries (e.g., the short-/long-term negative effects of lockdowns) and discuss variation in outcomes (if such exists). 

AUTHOR RESPONSE:

There have been a lot of variation between lockdowns in terms of the measures imposed, and also in the study designs to describe the impacts of lockdown. In the second paragraph of the discussion we outline some of these differences described by a meta-analysis of psychological health in COVID-19 induced lockdowns (reference 19).

A penultimate paragraph has been added to the Discussion section briefly covering the points raised by the reviewer including context of a paper [32] describing changes in physical health post-lockdown for a specific group of people experiencing anxiety post-lockdown.

REVIEWER 5:

REVIEWER COMMENT:

Overall objective and the extent to which this was achieved:

This paper attempts to address a very important and, to date, a sparsely researched topic: When lockdown ceases, to what extent do the established negative health effects of lockdown persist and what is the speed of recovery of individuals and communities to these negative effects?

The authors state that they took advantage of a natural experiment where, contemporaneously, there was a comparison group outside the lockdown area in Victoria, Australia. Their stated aim was to examine (lines 83-87):

1. the impact on mental and physical health of the extended lockdown.

2. the impact of the extended lockdown on established determinants of health: unemployment, social interaction, and finances.

3. Effects persisting after lockdown was lifted.

Of these three aims, the literature is flooded with the impact of lockdowns on health, mental, physical, and psychological. This is also mentioned in their introduction. However, it is the third aspect that is under-researched as again, they explain (lines 70 -73) even before the three objectives are listed.

Main takeaway: Unfortunately, it is in this third objective that the paper falls short. In their conclusion, they mainly talk about the negative impact of the extended lockdown (not new ground) and the mitigating effect of employment and social interactions which also has been shown. They end by saying “longer-term follow-up is required to identify any persistent health effects of community lockdowns.” They actually could have done this had they exploited the 2 months of post-lockdown data. But that was not done. 

AUTHOR RESPONSE:

This manuscript does describe outcomes post-lockdown and we have now made this clearer in the paper.

For example, Table 2 refers to outcomes during a ‘post-lockdown’ period, and a test labelled as ‘After lockdown’ describes differences between the exposure group and control group post-lockdown whilst accounting for differences between group pre-lockdown. 

Addressing point 3, our findings describe that after the conclusion of lockdown there were no persistent effects based upon the measures described in this paper. As other reviewers have pointed out, our findings do not evaluate some forms of mental health disorders such as PTSD which may develop after longer follow-up periods, or upon repeated exposure to lockdowns. 

Further discussion has been added on this point in the penultimate paragraph of the Discussion section.

The following text has been added to the Conclusion section :” After the removal of lockdown restrictions, the mental health of those exposed to the extended lockdown returned to be equivalent to those in the non-exposed comparison group”.

REVIEWER COMMENT:

Clarification needed: The paper seems remarkably similar to this already published:

Griffiths, D., Sheehan, L., Vreden, C.V., Petrie, D., Sim, M. and Collie, A., 2021. 1360 Health impacts of a 4 month community-wide lockdown: a prospective longitudinal study in Victoria, Australia. International Journal of Epidemiology, 50(Supplement_1), pp.dyab168-240. Though this appears in Google Scholar, do I assume that this is just the publication of an abstract of a conference presentation? I was unable to get hold of this paper. The authors can clarify.

AUTHOR RESPONSE:

The editor has been informed this is an abstract for an oral conference presentation.

REVIEWER COMMENT:

• The Model:

1. I failed to find out what is the model that they are estimating. Lines 170 to 189 give a general description of what was done. The exact specification of the model/models needs to be spelled out, and the control variables indicated. This is the heart of the paper and a general description does not provide the information needed to assess the underpinnings of the analysis 

AUTHOR RESPONSE:

Changes have been made to the text to clarify the analytical approach.

Figures 1 and 2 have updated y-axis labels describing the scales used to measure health outcomes.

Fixed effects for time and location have been emphasised for mixed linear regression models (for health outcomes) in addition to describing the exposure.

Methods>Analytical approach:

“Fixed effects were included for gender, age group, survey mode, residential location and survey timepoint.”

Main effects for time and location have been emphasised for generalised estimating equations (for outcomes describing known determinants of health)

Methods>Analytical approach:

“Main effects were included for gender, age group, survey mode, residential location and survey timepoint.

An additional supplementary table S2 has been added describing the exposure group and control group across a set of covariates.

REVIEWER COMMENT:

2. In the estimation, they also need to add a fixed time effect. 

AUTHOR RESPONSE:

We have included a fixed time effect in each model as the exposure and have added text to the Analytic Approach section to emphasise this point. This is described in the previous response.

REVIEWER COMMENT:

Variables:

3. What were all the variables and how were they defined and measured? For instance, I could find no definition of the two key variables they have used and even shown graphs with them: psychological distress and mental health. 

AUTHOR RESPONSE:

The y-axis for Figures 1 and 2 have been renamed to reflect the description of the Kessler 6 score and 2-item Short Form health survey (SF-12) mental, or physical, component scores. 

This matches the descriptions listed in the Methods>Health outcomes subsection.

REVIEWER COMMENT:

4. Of the variables they have discussed, I have the following comments:

i) Financial stress is important to capture. However, the authors have taken only one out of the 3 indicators that the GSS Summary Results for Australia have considered. The GSS considers three elements:

(https://www.abs.gov.au/statistics/people/people-and-communities/general-social-survey-summary-results-australia/2014) a. A cash flow problem in the last year in terms of either inability to pay bills on time or having to get help to pay.

b. The ability to raise $2000 within a week for something important.

c. Taking a dissaving action such as “drawing on savings, increasing a credit card balance by $1,000 or more and taking out a personal loan”.

Please see Graph 2 in https://www.abs.gov.au/statistics/people/people-and-communities/general-social-survey-summary-results-australia/2014. It can be seen that of all the 3 stressors, cash flow and dissaving have higher percentages in general than the ability to raise $2000 dollars. In fact, it is the unemployed that have the most problems. The authors need to justify why the other two were not considered and why focusing on the selected item was appropriate.

AUTHOR RESPONSE:

We acknowledge that the analysis includes the question listed by the reviewer as point b without the other two questions listed. 

There are two indicators described by the reviewer. Both are in reference to a 12-month period of time, which are not suitable for shorter-term changes in financial stress. Our approach measures changes across three time-points within a 6 month period. Thus, the question on “The ability to raise $2000 within a week for something important” is most relevant for our purpose.

The approach of taking point b alone has been published previously (below).

Example publications using the financial resources question without the others from the General Social Survey:

Pollock D, Shepherd CC, Adane AA, Foord C, Farrant BM, Warland J. Knowing your audience: Investigating stillbirth knowledge and perceptions in the general population to inform future public health campaigns. Women and Birth. 2021 Jul 30.

Thomas DP, Briggs V, Anderson IP, Cunningham J. The social determinants of being an Indigenous non‐smoker. Australian and New Zealand journal of public health. 2008 Apr;32(2):110-6.

Griffiths D, Sheehan L, van Vreden C et al. The Impact of Work Loss on Mental and Physical Health During the COVID-19 Pandemic: Baseline Findings from a Prospective Cohort Study. J Occup Rehabil. 2021. doi:10.1007/s10926-021-09958-7

REVIEWER COMMENT:

ii) I do not understand how, (if the responses to that one question was taken as “yes” or “no” or “don’t know”), financial stress could be measured (for each person) on a 1 to 10 scale. Is this an overall (average) measure for the county? This needs clarification.

AUTHOR RESPONSE:

We report two measures described in the final 3 sentences of the Methods>Determinants of health subsection

(1) Financial resources 

Question:‘If all of a sudden you had to get $2000 for something important, could the money be obtained within a week?

Response: Yes, No or Don’t know

(2) Financial stress (current level)

Response: a 10-point scale from 1 (not at all stressed) to 10 (as stressed as can be)

These are two different measures, which are described in the final 3 sentences of the Methods>Determinants of health subsection

REVIEWER COMMENT:

5. Social Interaction – two binary measures are taken:

Was any time spent with anyone not a household member and

Whether contact was made with 7 or fewer people via speech, phone or online communication.

The original 35 item Duke Social Support Index was abbreviated to a 23-item index and also to a 11-item index for use on elderly patients who may get exhausted by the use of the longer questionnaire. However, studies have generally used either the 23 item or the 11 item DSSI. Usually these have been for older people. In this paper (not confined only to the elderly), only two items have been picked and the justification of these two items has not been given. In other words, why would we believe in the validity of a 2-item DSSI?

The authors need to justify and discuss this. 

AUTHOR RESPONSE:

We have added justification for the inclusion of 2 questions that are used to derive the Social Interaction sub-scale of the Duke Social Support Index. 

The following text has been added in Methods>Determinants of health:

“The social interaction sub-scale itself was not derived following participant feedback on the question regarding taking part in group meetings (such as sports or other clubs). Many of these activities were not legally permitted under lockdown restrictions and therefore if included the overall interaction sub-scale would instead partially reflect compliance with lockdown restrictions “

REVIEWER COMMENT:

6. What is the need for the variable “engagement in work” since “employment” at time of survey is also included? Is there any extra information being captured? 

AUTHOR RESPONSE:

Public health measures to reduce viral transmission during the pandemic resulted in a large number of people either losing their job or being temporarily stood-down from work (but remaining employed). 

The two variables referred to as ‘employment’ and ‘engagement in work’ reflect this difference, which is also reflected in the study results. Many people were employed but temporarily not engaged in work. 

REVIEWER COMMENT:

• Other comments regarding results:

Turning to the results with regard to the stated objectives:

7. The negative health findings (they focus on psychological distress and mental health --- again, how were these measured?) during lockdown for both exposed and comparison group is in line with findings elsewhere in the COVID literature and so do not break new ground..

AUTHOR RESPONSE:

There is a section Methods>Health Outcomes that describes how health was measured using the Kessler-6 psychological distress scale and scores of mental and physical health from the 12-item short form health survey.

The y-axes for Figures 1 and 2 have been updated to assist the reader identify the scales upon which health outcomes are presented.

We also examine this post-lockdown.

REVIEWER COMMENT:

Unobserved heterogeneities as the two groups do not appear similar

8. During the lockdown period the mental health of the comparison group actually improved significantly from pre-lockdown. There is no further exploration of this counter-intuitive result. However, this pattern did not show up in the exposure group thus leading one to suspect that there are other factors at play or unobserved heterogeneities that have not been captured and which may, thus, invalidate this “natural experiment”. 

AUTHOR RESPONSE:

The pre-lockdown setting took place during the early stages of the pandemic. We have now made this clearer in the paper. Also in the final sentence the subsection Methods>Participants date and context we describe significant shifts in health from pre-pandemic levels. 

Given the significant rapid declines in mental health at the start of the pandemic (pre-lockdown), it is perhaps unsurprising that mental health would recover over time in the rest of Australia (having relatively few restrictions). 

We didn’t observe any significant differences post-lockdown between the group exposed to lockdown and the control group, suggesting a similar pattern of recovery in mental health, which was delayed for those experiencing lockdown.

This is described in: Discussion>paragraph 1 

“Following an abrupt national decline in mental health observed during the early stages of the pandemic, people exposed to an extended lockdown experienced a delay in the recovery of mental health observed in the rest of Australia. Following the easing of lockdown measures, we then observe a resolution of the negative mental health impacts of lockdown within a two-month period, bringing the mental health of Victorian residents back in line with that of the rest of Australia.”

REVIEWER COMMENT:

9. Right now, looking at the mental health and psychological graphs, it can be seen that the two groups did not begin at the same level. Are these differences statistically significant? This is important as otherwise, this further shows that the two groups were quite different to begin with (another reason to suspect other unaccounted for factors at play). 

AUTHOR RESPONSE:

Firstly, we graphically present averages for the exposure group and comparison group across three time-points which are unadjusted, or raw data.

However, in the regression models we account for differences between the study groups by incorporating covariates into the models. After adjusting for these covariates there are no significant differences between the two groups at the first survey time-point (Table 2).

The part of Table 2 labelled ‘The Lockdown Impact’ involves another form of adjustment. It models differences at one time point whilst controlling for differences at another time point.

In summary the analytical approach accounts for the concerns of the reviewer observed from the raw averages.

REVIEWER COMMENT:

10. What accounts for the different trajectories of psychological distress? Again, the two groups appear different.. 

AUTHOR RESPONSE:

This paper models changes in health over time starting just after the first wave for an exposure group that experienced an extended community lockdown due to a second wave, and a comparison group that did not experience a community lockdown. 

We conclude that some differences in psychological distress coincide with a community lockdown but these appear to resolve quickly after the lockdown ended. 

Additionally, there are trends describing a recovery in mental health over time in general towards pre-pandemic population levels.

REVIEWER COMMENT:

• The graphs

11. The graphs are not easy to understand.

i. Since there is no discussion of “health score” in the cases of mental health and psychological distress, one does not know how to interpret the y-axes. The x-axis is not labelled. 

AUTHOR RESPONSE:

Figure 1 and Figure 2 have been updated. 

The y-axes are labelled according to the specific scale from which the values are taken. 

The x-axis is now labelled as time. 

REVIEWER COMMENT:

ii. How were the health scores obtained? 

AUTHOR RESPONSE:

Health scores are described in the short subsection Methods>Health outcomes. The updated y-axis on Figures also act to emphasise the nature of the scales.

REVIEWER COMMENT:

iii. As I understand, the first dot is for data collected PRIOR to lockdown. The second dot is the data collected DURING lockdown. The third dot is the data collected POST lockdown. It seems a valuable opportunity to explore the POST lockdown period was not taken by averaging the October to December data. This would have shown the speed of recovery and might have added some new information to the literature. 

AUTHOR RESPONSE:

We agree that exploring time-dependent differences in the post-lockdown time-point would be interesting for the exposed group, but this was beyond the scope of this paper.

REVIEWER COMMENT:

• Precision of estimates and statistical power:

12. Since I am not clear on the models estimated to find the mental and physical effects of a lockdown, I wanted to make sure that if any of the explanatory variables were correlated, that was clearly brought out and the implications mentioned. Employment, social interaction, and financial stress are likely to be highly correlated. Though there are methods to deal with this problem, at least the impact on the precision of the estimates needs to be mentioned. 

AUTHOR RESPONSE:

Employment, social interaction and financial stress (along with mental and physical health) are defined as outcomes in the regression models and are not treated as explanatory variables. 

This information can be found in

Methods>Analytical approach:

“Generalised estimating equations were used with binary logistic models for binary outcomes describing work status, social isolation, and financial resources”

Whilst we observe that changes in mental health coincide with social isolation and loss of work during lockdown, these are each treated as separate outcomes for separate models. This approach is aligned to the research questions. 

REVIEWER COMMENT:

13. The state of Victoria has 6.68 million people (line 94) accounting for over a quarter of the 6.68 million population of Australia (line 95). I find it worrying that the analysis has only 305 people in Victoria (exposed group) and 593 in the rest of Australia (the comparison group). The sample sizes seem too small for the statistical tests to have much power. At the very least, this should be mentioned. 

AUTHOR RESPONSE:

The sample sizes were sufficient to identify significant differences in outcomes during the lockdown (adjusted estimates reported in Table 2). And if these effects had persisted at the same level after the lockdown we would have had enough statistical power to detect these. Following the same individuals over time also provides us with more statistical power when estimating the change in outcomes as outcomes are correlated within individuals.

Within the discussion, we acknowledge the limitation of using a cohort sample, compared with population level data, and have added a remark in alignment with the reviewers comment in regard to financial stress:

Discussion:

 “Furthermore, we did not identify any significant differences in financial stress due to the lockdown within our sample”

---

## [Decision Letter · Decision Letter 1]

14 Mar 2022

PONE-D-21-24391R1The health impacts of a 4-month long community-wide COVID-19 lockdown: Findings from a prospective longitudinal study in the state of Victoria, Australia.PLOS ONE

Dear Dr. Griffiths,

Thank you for submitting your manuscript to PLOS ONE. After careful consideration, we feel that it has merit but does not fully meet PLOS ONE’s publication criteria as it currently stands. Therefore, we invite you to submit a revised version of the manuscript that addresses the points raised during the review process.

We look forward to receiving your revised manuscript.

Kind regards,

Giuseppe Carrà, PhD

Academic Editor

PLOS ONE

Journal Requirements:

Reviewers' comments:

Reviewer's Responses to Questions

**Comments to the Author**

1. If the authors have adequately addressed your comments raised in a previous round of review and you feel that this manuscript is now acceptable for publication, you may indicate that here to bypass the “Comments to the Author” section, enter your conflict of interest statement in the “Confidential to Editor” section, and submit your "Accept" recommendation.

Reviewer #1: All comments have been addressed

Reviewer #2: (No Response)

Reviewer #4: All comments have been addressed

2. Is the manuscript technically sound, and do the data support the conclusions?

Reviewer #1: Yes

Reviewer #2: Partly

Reviewer #4: Yes

3. Has the statistical analysis been performed appropriately and rigorously? 

Reviewer #1: Yes

Reviewer #2: I Don't Know

Reviewer #4: Yes

4. Have the authors made all data underlying the findings in their manuscript fully available?

Reviewer #1: No

Reviewer #2: Yes

Reviewer #4: Yes

5. Is the manuscript presented in an intelligible fashion and written in standard English?

Reviewer #1: Yes

Reviewer #2: Yes

Reviewer #4: Yes

6. Review Comments to the Author

Reviewer #1: (No Response)

Reviewer #2: The authors acknowledge that their methodology does not allow the capture of data on the most relevant possible mental health impact — post-traumatic stress disorder (PTSD). This condition has a clear temporal element: onset of symptoms is often delayed until some time after a traumatising event. There is popular media discussion indicating mental health practitioners and the community itself are reckoning with potentially traumatic effects of lockdown in combination with alarm caused by the pandemic itself (e.g. ABC 2021; Coslett 2021; Sarner 2021; Watson 2021). Yet this paper repeatedly claims to demonstrate that mental health impacts did not persist following the conclusion of lockdown, a conclusion framed by the research questions as posed on lines 83-87 of the manuscript.

The authors used instruments (K-6, SF-12) that are not validated for the detection of PTSD. Although they measure symptoms that may accompany PTSD, these symptoms may have taken time to emerge following the cessation of lockdown. The study does not include data from any subsequent time points that could be used to assess whether there have been delayed effects from lockdown. The paper does not acknowledge the limited scope of its exploration of mental health until line 385: ‘Whilst our findings describe a recovery in aspects of mental health such as symptoms of anxiety and psychological distress…’ That rather more circumspect framing should be used throughout the entire paper, instead of referring to ‘mental health’ more generally.

REFERENCES

ABC 2021 https://www.abc.net.au/radio/melbourne/programs/theconversationhour/the-conversation-hour/13362804

Coslett 2021 https://www.theguardian.com/commentisfree/2021/dec/07/covid-ptsd-cases-mental-health-crisis-early-intervention-trauma-uk

Sarner 2021 https://www.theguardian.com/lifeandstyle/2021/apr/14/brain-fog-how-trauma-uncertainty-and-isolation-have-affected-our-minds-and-memory

Watson 2021 https://www.abc.net.au/everyday/lockdown-is-harder-than-before-trauma-specialist-explains-why/100252974

Reviewer #4: (No Response)

7. PLOS authors have the option to publish the peer review history of their article (what does this mean?). If published, this will include your full peer review and any attached files.

Reviewer #1: No

Reviewer #2: No

Reviewer #4: No

---

## [Author Response · Author response to Decision Letter 1]

18 Mar 2022

Reviewer 2 did not respond provide a response on the question of whether comments were addressed, and provided a comment that our analysis does not describe PTSD.

AUTHOR RESPONSE TO REVIEWER 2 COMMENT:

The study objectives do not propose to measure post-traumatic stress disorder (PTSD) but rather it is designed to measure mental health generally – Mental Health as a person’s emotional, cognitive and intellectual status. It could be considered positive or negative at a given time.

Mental illness, or psychological disorders such as PTSD, is a different construct which can affect thinking, feeling, mood, behaviour and ability to function over time – we don’t propose to describe this outcome in our study objectives.

There is a difference between mental health and mental illness/conditions/disorders, although these can often be used interchangeably.

We claim to describe mental health using the SF-12 score (on a numeric scale) – an approach which is appropriate for this construct, validated, and well-defined in the Methods section. 

We also describe a measure of overall physical health - analogously this does not describe specific types of physical impairments, or types of injury for example.

The reviewer states that the methodology does not directly capture post-traumatic stress disorders, yet the reviewer also describes that this is already acknowledged in the study limitations of the paper, citing part of the following sentence:

From Line 385:

“Whilst our findings describe a recovery in aspects of mental health such as symptoms of anxiety and psychological distress, mental disorders such as post-traumatic stress disorder may re-emerge over longer periods of time [33], or upon repeated exposure to lockdowns or crises.”

We do not propose to make claims about PTSD and we also acknowledge the limitation of the measures used in not being able to provide information about specific mental illnesses such as PTSD, Depression or Anxiety. 

No changes to the text were made.

---

## [Editor Report · Decision Letter 2]

25 Mar 2022

The health impacts of a 4-month long community-wide COVID-19 lockdown: Findings from a prospective longitudinal study in the state of Victoria, Australia.

PONE-D-21-24391R2

Dear Dr. Griffiths,

We’re pleased to inform you that your manuscript has been judged scientifically suitable for publication and will be formally accepted for publication once it meets all outstanding technical requirements.

Kind regards,

Giuseppe Carrà, PhD

Academic Editor

PLOS ONE

---

## [Editor Report · Acceptance letter]

31 Mar 2022

PONE-D-21-24391R2 

The health impacts of a 4-month long community-wide COVID-19 lockdown: Findings from a prospective longitudinal study in the state of Victoria, Australia. 

Dear Dr. Griffiths:

I'm pleased to inform you that your manuscript has been deemed suitable for publication in PLOS ONE. Congratulations! Your manuscript is now with our production department. 

Kind regards, 

on behalf of

Dr. Giuseppe Carrà 

Academic Editor

PLOS ONE